# Microstructural, Physicochemical, Microbiological, and Organoleptic Characteristics of Sugar- and Fat-Free Ice Cream from Buffalo Milk

**DOI:** 10.3390/foods11030490

**Published:** 2022-02-08

**Authors:** Atallah A. Atallah, Osama M. Morsy, Wael Abbas, El-Sayed G. Khater

**Affiliations:** 1Department of Dairy Science, Faculty of Agriculture, Benha University, Toukh P.O. Box 13736, Egypt; 2Department of Basic and Applied Sciences, Arab Academy for Science, Technology and Maritime Transport (AASTMT), Cairo P.O. Box 2033, Egypt; ossama.morsy22@gmail.com (O.M.M.); wael_abass@aast.edu (W.A.); 3Department of Agricultural and Biosystems Engineering, Faculty of Agriculture, Benha University, Toukh P.O. Box 13736, Egypt; alsayed.khater@fagr.bu.edu.eg

**Keywords:** ice cream, sweeteners, quality characteristics, calorific value, sugar profiles, scanning electron microscopy

## Abstract

Ice cream is a popular dessert product across the world. Structure, body, taste, and odor properties are created by adding non-milk ingredients and milk ingredients. The main aim of the study is to decrease the caloric value of ice cream by using sugar and fat replacements. Ice cream treatments were investigated based on microstructural, chemical, physical, microbiological, sensory, and calorific values. Four different ice creams were used (control ice cream (SC1), ice cream with stevia (SC2), ice cream with sucralose (SC3), and ice cream with sorbitol (SC4)). The chemical properties in all treatments of ice cream were significantly recorded (*p* < 0.05). The highest sucrose and fat levels were detected in the SC1 treatment compared with the other treatments (*p* < 0.05). The lowest fat and sugar amounts were observed in the SC2, SC3, and SC4 treatments (*p* < 0.05). The highest viscosity, overrun, and hardness values (*p* < 0.05) were detected in the control ice cream. Total aerobic mesophilic bacterial counts were not significantly recorded between different ice cream treatments (*p* < 0.05). The sensory scores were not significantly affected by sweeteners and bulk agents in the different treatments. The highest calorific value was calculated in the SC1 samples (*p* < 0.05). On the other hand, the lowest calorific value was calculated in SC2, followed by the SC3 and SC4 treatments. In scanning electron microscopy (SEM), the gel exhibited a homogeneous structure with a fine network within the SC2, SC3, and SC4 treatments, as it contained a cohesive structure with small-sized pores.

## 1. Introduction

Ice creams are a rich source of fat, carbohydrates, and protein, which reflects its energy value. The energy value of standard ice cream is almost 200 kcal/100 g. The consumer awareness of functional and healthier dairy has led to the development of new methods to produce ice cream [1,2]. Sucrose levels in ice cream range from 9% to 28%, and fat levels between 3% and 15%, of the total ingredients [3]. Ice cream products have a large nutritional value and are highly consumed by different age groups. Due to increasing consumer demand for low-energy dairy and foods, dairy products produced with sweetener and fat replacements have become highly popular recently [4]. Other dietary restrictions for the food industry include consumers with obesity, metabolic syndrome, diabetes, cardiovascular issues, and allergies, and those looking for low-sugar, low-fat, and low-calorie products. The WHO has determined the existence of a world-wide upsurge in diabetic cases, expected to rise by 57.2 million till 2025, in sharp contrast to the 19.4 million diabetic cases in 1995 [5,6]. High-intensity sweeteners stand out as an alternative to sucrose. Since they give no energy value to food products, they are preferred for those looking for low-sugar, low-fat, and low-calorie products, including diabetics and obese individuals [7]. The reduction of sucrose or fat to produce sucrose- and fat-free ice cream products affects the ice cream’s solids, and its freezing point depresses. The first can be compensated by adding a bulking agent such as maltodextrin or polydextrose, and the latter by adding a freezing point depressant such as sorbitol [8]. The successful production of high-quality, nutritional dairy products depends to a large extent on the imitation and simulation of the sensory characteristics of equivalent standard products. Consequently, the development of a completely new treatment is often required to obtain acceptable dietetic ice cream for consumers. Carbohydrate-based filler agents (maltodextrin and polydextrose) have recently been used for low-calorie processing because they have the least negative impact on the production, shelf life, and price of ice cream [9].

Sugar gives a sweet and pure taste to ice cream and thus has a crucial regulating effect on the overall taste sensation. To produce ice cream, which is sufficiently low in calories, as well as to provide a refreshing frozen dessert for obese and diabetic individuals, it is necessary to reduce or remove the sugar content. Once fat levels are between 3% and 15% and sucrose levels between 9% and 28%, the sugar and fat are to be replaced by intense sweeteners and bulking agents. Hence, there are some expected problems (recompense of the total solids to inhibit weak texture and body). This can usually be achieved using low-calorie bulking agents (i.e., polydextrose and maltodextrin). However, the addition of bulking agents creates somewhat smooth products because they lower the freezing point to a much lower degree than sucrose does. Bulking agents impart creaminess and smoothness improves texture, and thus they provide a mouthfeel and protection against temperature fluctuation to please customers [10].

Sweeteners can be classified as natural or artificial. Stevia powder (*Stevia rebaudiana*) is a natural sweetener and is about 250–300 times sweeter than sucrose, depending on the dispersion matrix [11]. It is a very low-calorie compound (one gram gives zero calories), which makes it a good alternative to sugar for patients suffering from diabetes mellitus and other sedentary-life-related diseases [12]. Stevia is recognized as safe supplement by the JECFA, WHO, and FDA, with relatively high upper limits [13,14,15]. Sucralose is an artificial sweetener and is about 600 times sweeter than sucrose. It is a calorie compound (one gram gives 3.36 calories), which makes it a good alternative to sugar [16]. Sorbitols containing sugar-free products have a lower glycemic index and are about 60% more sweet than those products containing sucrose (one gram gives 2.6 calories). The use of these sweeteners in food is very useful as they impart sweetness without adding sugar, which results in reduced calories and helps in weight loss and diet control. Artificial sweeteners are considered safe as some of them, such as sucralose, are not digested by our bodies [17].

Ice cream is frozen dairy product that includes healthy and nutritious aspects. Ice cream is a rich source of sugar, protein, and fat. The sucrose level in ice cream products changes between 9% and 30% and the fat content ranges from 2% to 15% of the total components. Due to the high prevalence of Type 2 diabetes and obesity among children and adolescents, people are now more aware of their health status, and hence conscious of their diet. This health-conscious decision poses a formidable challenge to ice cream production. As a result, the ice cream market trend is moving towards a sugar- and fat-free ice cream formulation with excellent texture, structure, and sensory attributes to gain consumer satisfaction. The aim of this study is to prepare a low-calorific value ice cream using different sweeteners and bulking agents as alternatives for sucrose and fat, and to compare their potential influences on the chemical, physical, microbiological, and sensory properties, as well as calorific value, of the ice cream.

## 2. Materials and Methods

### 2.1. Materials

Fresh, skimmed buffalo milk and buffalo cream (35% fat) were provided by the Dairy Science Department of the Faculty of Agriculture, Moshtohor, Benha University, Qalubia, Egypt. Skimmed bovine milk powder (extra grade, spray dried) was obtained from California Dairies, Inc., Fresno, CA, USA. Sodium carboxy methyl cellulose (CMC) was provided by the El-Nasr Company for Chemicals, Cairo, Egypt. Vanilla was obtained from the Tag El-Melouk Company for Food Industries, 6th of October City, Egypt. Commercial grade granulated sugar cane was obtained from the Egyptian Sugar and Integrated Industries Company, Hawmdia, Giza, Egypt. Sorbitol (C_6_H_14_O_6_) was provided by Techno pharmchem, Bahdurarh Company, Jhajjar, India. Stevia powder (*Stevia rebaudiana*) was obtained from the Techno pharmchem, Bahdurarh Company, Jhajjar, India. Sucralose was obtained from Tale and Lyle specialist sweeteners, London, UK. Maltodextrin was provided by Heilongjiano Haotian Corn Development Co., Ltd., Shihua, China. Polydextrose was obtained from Dalya Foreign Trade Co., Ltd., Istanbul, Turkey.

### 2.2. Ice Cream Preparation

The ice cream was prepared according to Whelan et al. [18]. The base ice cream mix was standardized to contain 6% fat, 12% solid not-fat, 15% sugar, and 0.25% stabilizer. The calculated amount of fresh, skimmed buffalo milk was heated to 40 °C. Solid materials such as skimmed bovine milk powder, sugar or sweetener, stabilizer, and fresh buffalo cream were incorporated with continuous agitation to produce the base ice cream mix. The mix was heated for 10 min at 80 °C, then immediately cooled to 5 ± 1 °C and aged for approximately 24 h at 5 ± 1 °C. After that, different treatments of ice cream mixtures were mixed with vanilla (0.01%). Different mixes were whipped during freezing in a horizontal batch freezer (CARPIGIANI machine, Anzola dell’Emilia, Italy). The resultant ice cream was packaged in sterilized plastic cups (50 mL and 100 mL), hardened at −26 °C for 24 h in a deep freezer, and then kept at −20 °C. All samples were analyzed for chemical, physical, microbiological, and sensory properties, and calorific values of the ice cream mixes are shown in Table 1.

Initial experiments were conducted to make ice cream using different amounts of stevia, sorbitol, and sucralose. The best values of stevia (0.06%), sucralose (0.03%), and sorbitol (3%) in the ice cream mixes were determined by performing tests such as sensory evaluation and physical tests. As well, initial tests were conducted to prepare ice cream using different values of maltodextrin and polydextrose as bulking agents. The best value of maltodextrin was 7.5%, and polydextrose, 7.5%, in the produced ice cream (results not shown).

### 2.3. Chemical Analysis

The total solids (TS) of all groups were determined by drying 2 to 3 g at 105 °C using an air oven (Thermalize Scientific, New York, NY, USA) until constant weight was achieved [19]. The total protein (TP) levels of all samples were calculated by the Kjeldahl method according to IDF [20], as follows:TP = TN × 6.38(1)
where TP is the total protein and TN is the total nitrogen.

Fat values were calculated by ISO [21], applying the Gerber method. Ash values were detected by drying the samples at 100–105 °C, and by combustion of solid samples at 550 °C in an electric muffle furnace (Protherm PLF 110/15, Alserteknik A.S., Ankara, Turkey) [22]. Titratable acidity level was measured by titration with NaOH (0.1 N) using phenolphthalein (Oxoid, 0.1%) [23].

### 2.4. Sugar Profile Analysis

The sugar patterns were observed according to Arslaner et al. [24]. A five-gram sample was diluted with 20 mL of a water–methanol mix (75:25; *v*/*v*) and then centrifuged at 5000 *g* for 10 min at 5 °C. The supernatant was filtered using Whatman No. 1, and then membrane filtered (0.45 µm, GE Healthcare Life Sciences, Chicago, IL, USA). Previous extracts were transferred to 2 mL vials and stored at −20 °C until used. The sugar profiles were measured by high-performance liquid chromatography equipment [(HPLC), LC-10A Series; Shimadzu, Japan], a refractive-index detector (RID-10A), an LC-16ADVP binary pump, a DCou-14A, a guard column (Sc-LcShodex), and an acetonitrile–water mix (as the solvent (80:20; *v*/*v*); 2 mL/min flow rate). 20 µL of the extract was injected in the column temperature at 40 °C. Sugars (glucose, fructose, galactose, sucrose, and lactose) were identified by comparing their sugar standards with retention times.

### 2.5. Microbiological Properties

The total aerobic mesophilic bacterial count was carried out in a plate count agar (Oxoid) incubated at 30 °C for 48 h, and at 5 °C for 10 days for total psychrotrophic bacteria [25]. Coliform bacteria were observed by APHA [26]. The counts of yeast and mold were investigated as mentioned by IDF [27]. The counts were calculated as logarithm colony-forming-units per gram (log_10_ CFU g^−1^), and in bacteria were enumerated from 30 to 300 colonies, and yeasts and molds were counted from 15 to 150 colonies of samples.

### 2.6. Physical Analysis

The freezing point was determined according to Marshall et al. [28], applying the digital thermometer. The freezing point of ice cream was measured as soon as it exited the ice cream machine by placing the thermometer directly in the treatment that came out of the machine and then recording the reading. The viscosity (cP) of mixes was detected according to the Brookfield measurement at 5 °C using spindle No. #07 for 50 rpm, then the reading was recorded after 30 s in 250 mL cups of ice cream mixes. Overrun of ice cream samples was calculated by using the method given by Akin et al. [29] as follows:Overrun = [(A − B)/B] × 100(2)
where A is the weight of a volume of the mix and B is the weight of the same volume of ice cream.

The weight of melted ice cream expressed as a percentage of the initial weight of ice cream was measured according to Arndt and Wehling [30]. Ice cream samples weighing 100 g were placed into wire mesh (6 pores/cm^2^) over a glass funnel fitted on a conical flask at ambient temperature. The melted ice cream was weighed after 30 min.

### 2.7. Hardness

The hardness of ice cream samples was determined by a Universal Testing Machine (Texture Pro^TM^, Tokyo, Japan) equipped with a load cell (250 Ibf) and connected to a computer programmed with Texture Pro^TM^ texture analysis software (Texture Pro^TM^, program, DEVTPA with the hold). A flat rod probe (49.95 mm in diameter) was used to uniaxially compress the ice cream samples to 50% of their original height. The hardness was adjusted to a test speed of 60 mm/s, with a trigger force of 1 N, deformation of 25%, and a holding time of 2 s between cycles at −20 °C. 

### 2.8. Sensory Analysis

Sensory evaluation of the ice cream was measured according to Khalil and Blassy [31], was performed after getting consent of the dairy science department’s proving committee, Faculty of Agriculture, Benha University, Egypt. A panel composed of 15 members of the Dairy Science Department, Faculty of Agriculture, Benha University Egypt, was assembled. Panel members were selected based on their interest in the sensory evaluations of ice cream and were trained by testing commercial ice cream. Samples (50 g) were given to a group of 15 test panelists. The 15 experienced panelists ranged in age from 25–55 years. They evaluated the external color and appearance (10-point hedonic scale), structure and consistency (10-point hedonic scale), taste and odor (10-point hedonic scale), icy structure (10-point hedonic scale), melt in mouth (10-point hedonic scale), gummy structure (10-point hedonic scale), and total acceptability (10-point hedonic scale) for the samples. The samples were randomly coded and presented to each panelist at −20 °C.

### 2.9. Calorific Value

Calorific values of the ice cream samples were calculated according to the following:Calorific value = % sugar × 3.87 + % fat × 8.79 + protein × 4.27(3)

### 2.10. Scanning Electron Microscopy (SEM)

Morphological properties of the ice cream samples were determined using SEM at the National Research Centre, Giza, Egypt. Samples were prepared as described by Jaya [32] as follows: The samples were fixed on an iron stub and then made electrically conductive by coating them (in a vacuum chamber) with a thin layer of gold for 40 s. The moisture of samples was entirely removed by placing the samples in an air-tight desiccator containing silica gel. The weight of samples was periodically determined until achieving constant weight to confirm the complete removal of moisture. At least four images of typical structures at 6000 magnifications were recorded using an SEM (FEI Company, Eindhoven, The Netherlands), model quanta 250 FEG (field emission gun), attached with an EDX unit (energy dispersive X-ray analyses). The images were taken at an excitation voltage of 20 kV at different magnifications ranging from 400 to 6000, with working distance varying from 13.7 to 14.2 mm.

### 2.11. Statistical Analysis

Results were subjected to ANOVA and statistical variances were analyzed by Duncan’s test. Variances were considered significant at *p* < 0.05 and expressed as the mean and standard error (means ± SE). qPCR datasets were analyzed using PROC GLM of SAS [33]. The applied static model is as follows:Y_ij_ = µ + T_i_ + e_ij_(4)
where Y_ij_ is the dependent variable, µ is the overall mean, T_i_ is the effect of treatment (I = 1, …, 7), and e_ij_ is the residual standard error.

## 3. Results

### 3.1. Chemical Analysis

The chemical parameters of the ice cream treatments are shown in Table 2. The chemical parameters were significantly affected by the addition of sweeteners and bulking agents. The addition of sweeteners (stevia, sucralose, and sorbitol) and bulking agents had a significant (*p* < 0.05) effect on the change of total solids, fat, protein, and ash values of all treatments. The total solid changed between 27.98 ± 1.03% and 31.95 ± 1.54% (*w*/*w*) in all treatments. Protein content changed between 3.53 ± 0.19% and 3.97 ± 0.15% (*w*/*w*) in all groups. Fat value was 6.08 ± 0.10% (*w*/*w*) in the SC1 sample and was not detected in the other treatments. Ash content varied from 0.86 ± 0.01% to 1.01 ± 0.00% (*w*/*w*) in all samples. Titratable acidity values were not significantly varied between all ice cream treatments. The highest levels of total solids, fat, protein, and ash were detected in the control samples (SC1). The replacement of sucrose with sweeteners and bulking agents influenced the chemical properties of the ice cream mixtures.

### 3.2. Sugar Profiles

Table 3 shows the sugar pattern levels for all ice cream treatments. Replacement of sucrose with sweeteners and bulking agents had a significant (*p* < 0.05) effect on the change of glucose, fructose, sucrose, and maltose values in all groups. Glucose ranged from 0.20 ± 0.13% to 0.73 ± 0.10% in all samples. The highest glucose value was observed in the SC1 treatment compared to the other treatments. In the SC1 treatment, fructose and sucrose levels were 0.20 ± 0.07% and 14.97 ± 01.02%, respectively, while these values were not observed in the SC2, SC3, and SC4 treatments. The highest level of sucrose (14.97%) was obtained in the SC1 treatment, while sucrose not detected in the other treatments. The addition of sweeteners and bulking agents had no significant (*p* > 0.05) effect on the change of galactose and lactose contents in all treatments. Lactose values changed between 3.60 ± 0.21% and 3.76 ± 0.19% in all treatments. Generally, the replacement of sucrose with sweeteners caused a significant reduction in the sugars and calories in the ice cream treatments.

### 3.3. Physical Analyses

#### 3.3.1. Overrun

The overrun of ice cream is an important property since it has a direct relationship with the yield and profit. It affects the body, texture, and palatability of the ice cream. It is the incorporated amount of air in the mixtures of ice cream, producing foam which is stabilized by surface active agents. The overrun of ice cream samples is shown in Table 4. Overrun was significantly affected (*p* < 0.05) by sucrose replacement using sweeteners and bulking agents. Overrun changed between 57.95 ± 1.60% to 59.82 ± 1.56% in all samples. The overrun level in the SC1 treatment was higher than in the other treatments.

#### 3.3.2. Viscosity

Viscosity had been considered an important property of ice cream mixtures, and up to a certain extent, it appears essential for whipping ability and retention of air. The viscosity value of mixtures is affected by fat, protein, stabilizer, bulking agent, and the quality of materials used. Table 4 shows the viscosity levels of all samples. There were significant differences (*p* < 0.05) between all treatments. The viscosity levels ranged from 89.94 ± 4.56 to 92.57 ± 4.23 cP in all treatments. The highest levels of viscosity were recorded in the SC1 treatment, while the lowest levels were detected in the SC2, SC3, and SC4 treatments.

#### 3.3.3. Freezing Point

The freezing point values of the ice cream mix samples are presented in Table 4. The freezing point of ice cream ranged from −2.29 ± 0.15 °C to -3.35 ± 0.24 °C for all treatments. There were significant differences (*p* < 0.05) between all treatments. Levels of freezing point were affected by the replacement of sucrose with sweeteners and bulking agents.

#### 3.3.4. Hardness

The hardness of the products at the temperature at which it has the optimum consistency for dipping or scooping is an important consideration. Hardness is affected by many factors: total solids, overrun, principal melting point, amount, and type of stabilizer, etc. Therefore, it is desirable to have the overruns and melting points of all ice creams nearly the same. Melting and freezing points decreased as the level of water-soluble ingredients increased. The ice cream formulator must carefully choose the amounts and type of substances because they affect the melting and freezing points of the mixes. Table 4 shows the hardness values for all treatments. Hardness values ranged from 42.98 ± 3.90 N to 45.32 ± 3.56 N in all treatments. The results demonstrate significant differences (*p* < 0.05) between all treatments. The highest hardness value was detected in the SC1 treatment compared with the other treatments.

#### 3.3.5. Melting Values

Melting percentage is expressed as the loss of weight-tested sample compared to its initial weight. Additionally, melting percentage is a good indicator of the structure of the product. Figure 1 illustrates the variations of the melting percentage of the ice cream treatments. The results demonstrate that there were significant differences (*p* < 0.05) between all treatments. At zero time, melting percentage varied from 3.34 ± 1.24 g 100 g^−1^ to 4.21 ± 1.17 g 100 g^−1^ in all treatments. At 30 min, melting percentage changed between 20.56 ± 1.18 g 100 g^−1^ and 22.19 ± 1.08 g 100 g^−1^ in all samples. A lower melting percentage was detected in the SC1 treatment.

### 3.4. Microbiological Properties

Total aerobic mesophilic and psychrotrophic bacteria counts are presented in Table 5. Total aerobic mesophilic and psychrotrophic bacteria counts in all treatments were not significantly recorded (*p* > 0.05). The count of total aerobic mesophilic bacterial ranged from 4.10 ± 0.41 log_10_ CFU g^−1^ to 4.30 ± 0.54 log_10_ CFU g^−1^ for all treatments. The count of total psychrotrophic bacteria in all treatments was < 1 log_10_ CFU g^−1^. On the other hand, coliform bacteria, yeast, and mold counts were not detected in any of the ice cream treatments.

### 3.5. Sensory Analysis

The evaluation results for sensory characteristics (color and appearance, structure and consistency, taste and odor, icy structure, melt in mouth, gummy structure, and total acceptability) are presented in Figure 2. There were no significant differences (*p* > 0.05) in sensory scores for all ice cream treatments. The structure and consistency scores ranged from 9.80 ± 0.12 to 9.90 ± 0.18 in all samples. The taste and odor scores varied from 9.85 ± 0.10 to 9.90 ± 0.16 in all treatments. The melt in mouth scores varied from 9.70 ± 0.12 to 9.80 ± 0.18 in all treatments. Total acceptability scores ranged from 9.80 ± 0.07 to 9.90 ± 0.13 in all samples.

### 3.6. Energy Value

The caloric value of ice cream was calculated by taking the caloric value of carbohydrates, fat, and protein as 3.87 kcal g^−1^, 8.79 kcal g^−1^, and 4.27 kcal g^−1^, respectively (Table 6). There were significant differences (*p* < 0.05) in caloric values for all ice cream treatments. Total caloric values ranged from 68.35 kcal 100 g^−1^ to 148.60 kcal 100 g^−1^ in all treatments. The lowest caloric value was obtained in the SC2 treatment (68.35 ± 0.97 kcal 100 g^−1^), followed by the SC3 treatment (69.37 ± 0.85 kcal 100 g^−1^). The highest caloric value was observed in the SC1 treatment (148.60 ± 2.66 kcal 100 g^−1^).

### 3.7. Microstructure

The microstructures of ice cream samples are represented in Figure 3. This research shows the comparative effects of the sweeteners and bulking agents on ice cream microstructure by SEM. In the SEM images of the SC2, SC3, and SC4 treatments, the gel exhibited a homogeneous structure with a fine network. It contained a cohesive structure, with small-sized pores within the structure. The gel organization presented as regular, with long casein filaments. Chains of casein micelles were less apparent compared with the control samples (SC1). Further, the gel in the SC1 samples presented as irregular, with short and individualized casein filaments and higher clusters, and the gel appeared to have heterogenous pore sizes. The addition of maltodextrin and polydextrose led to a homogeneous microstructure with a fine matrix, obtaining very small pores. This microstructure would increase the numbers of bonds between particles and the dense and finely branched matrix in ice cream with maltodextrin and polydextrose.

## 4. Discussion

The main aim of the current study was to evaluate a novel fat- and sucrose-free ice cream using different sweeteners and bulking agents as alternatives for sucrose and fat, and to compare their potential influences on the chemical, physical, microbiological, and sensory properties, as well as the calorific values. The physical, textural, and sensory characteristics of ice cream are important quality properties because they play an important role in consumers’ acceptance of these products. Ice cream is a frozen dairy product that includes healthy and nutritious values. Ice cream is a rich source of sugar, protein, and fat. Sucrose levels in ice cream products range between 9% and 30% and fat content ranges from 2% to 15% of the total components. Due to the high prevalence of Type 2 diabetes and obesity among children and adolescents, people are now more aware of their health status, and hence conscious of their diet. This health-conscious decision poses a formidable challenge to ice cream production. The replacement of sucrose and fat with sweeteners and bulking agents in the preparation of ice cream can address the issues of current customers who center on normal and healthfully adjusted food sources [34]. Mainly, these innovative ice creams are prepared using artificial sweeteners and bulking agents.

With increased consumer attentiveness for efficient and improved dairy and foods, various new technologies have come to the fore to produce such products. Ice cream is one of the most served and loved frozen dairy products, but it is high in sugar levels (15%) and fat (6%); therefore, formulating its sugar- and fat-free version will serve as a good cause for decreasing the extra caloric intake and make it healthier. Stevia, sucralose, and sorbitol contained in sugar-free dairy products can aid in calorie reduction, weight loss, and diet control. Low-sugar dairy is important in dietary management as it allows the slow movement of glucose into blood, resulting in a very low rise in blood—glucose, obesity, and insulin levels [35].

Significant differences between the ice cream treatments in their chemical characteristics are due to the variance of the total solids, fat, ash, and protein levels in the mixtures. In a similar study, significant differences in the compositional properties were observed by Alizadeh et al. [35] when sucrose was replaced by stevia in ice cream. Deshmukh et al. [36] also found significant differences in the compositional characteristics of ice cream mixes when sucrose was replaced by a mix of stevia powder. Similar data were recorded by Ozdemir et al. [37], who reported a significant effect on the change of dry matter and sucrose values in all sugar-free ice cream samples. The highest levels of total solids, fat, and protein were found in the SC1 samples. In the SC1 samples, the highest values of sucrose, glucose, and fructose were detected. In the SC2, SC3, and SC4 treatments, sucrose and fructose values were not detected. This is due to the total replacement of sucrose and fat with sweeteners (stevia, sucralose, and sorbitol) and bulking agents. Considering that sucrose has many disadvantages due to its high glucose that facilitates the development of many metabolic diseases, such as diabetes mellitus, metabolic syndrome, and obesity [35], stevia-sweetened ice cream can be an alternative product for diabetic individuals. The lactose and galactose levels were not significantly shown in the ice cream samples. Since lactose is not completely (or just somewhat) digested in the small intestine system, it has a moderately low-calorie level, and along these lines, it tends to be useful to individuals who are sensitive to hyperglycemia [38]. The increasing occurrence of diabetes, obesity, and other health-related issues globally has led to the emergence of several healthy trends in dairy products in recent years. Consumers’ interest in healthy products has spurred innovation and led to the development of various healthy replacements for ingredients currently being used by the dairy industry [35].

The viscosity values saw progressive increases in the SC1 samples compared with the other treatments. This increase is due to the increase of the total solids, fat, sucrose, ash, and protein ratios in the mixtures. Alizadeh et al. [35] detected that the stevia addition decreased the viscosity values of the ice cream samples. As well, Ozdemir et al. [39] reported that the viscosity value of treatments containing sugar alcohols and HFCS was lower than that of the control samples. The highest value of overrun was obtained in the SC1 samples. This increase is due to the increase of the total solids, fat, sucrose, ash, and protein ratios in the mixtures, and the increase of the viscosity value. Alizadeh et al. [35] detected that the stevia addition decreased the overrun values of the ice cream samples, while the overrun was increased in the control samples. Further, Ozdemir et al. [39] reported that the overrun value of treatments containing sugar alcohols and HFCS was lower than that of the control samples. The freezing point values were affected by the replacement of sucrose with sweeteners. The lowest value of freezing point was detected in the SC1 samples. This decrease is due to the increase of the sucrose and total soluble solids ratios in the SC1 mixture. The highest hardness value was observed in the SC1 samples compared with the other samples. This increase is due to the increase of the total solids, fat, sucrose, ash, and protein ratios in the mixtures. Hardness levels in ice cream are influenced by factors such as overrun, ice crystal size, fat destabilization, ice phase volume, and the consistency properties of the mixture [40]. The highest melting percentage was detected in the SC2, SC3, and SC4 treatments. This might be related to the increase of freezing point value, which resulted in an increase of the required time for melting percentage. Further, this increase is due to the decrease of the total solids, fat, sucrose, ash, and protein ratios in the mixtures. In addition, they reported that sugars with lower molecular weight have a decreased melting percentage compared to those with higher molecular weight. Many factors can affect the melting percentage of ice cream samples, such as ice crystal size, fat destabilization, and the consistency coefficient of the mixture [40]. These data are confirmed by Pinto and Dharaiya [41], who found the frozen dessert melting percentage showed a decrease in the values of melting percentage that was detected for the SC1 treatment.

Aerobic mesophilic bacterial count for the ice cream showed no significant difference between the treatments. These results are similar to those of Arslaner et al. [24], who studied the effects of some sweeteners (sucrose, honey, and stevia) on the counts of standard plate in ice cream. They found that counts of standard plate were not statistically different between samples. Psychrotrophic bacteria in all ice cream treatments were <1 log_10_ CFU g^−1^. Coliform, yeast, and mold counts were not detected in any of the ice cream treatments. This may be due to the sufficient heat treatment of milk during manufacture and high sanitation conditions during the making of the ice cream. The authors of [17] found that the coliform, yeast, and mold count in low-fat, sugar free ice cream were absent. These results are similar to those of Arslaner et al. [24] for yogurt ice cream sweetened with sucrose, stevia, and honey.

The scores of taste and odor, structure and consistency, and total acceptability were not affected by the replacement of sugar with sweeteners and bulking agents for ice cream. This result confirms those of Verma [42], who reported an improvement in the sensory scores with the addition of maltodextrin with respect to flavor, body, texture, melting in mouth, and overall acceptability. The suitability of sensory scores was relatively higher in the replacement of sucrose with stevia and bulk agents for ice cream. Pinto and Dharaiya [41] detected the highest increase of sensory scores in low-fat, sugar-free frozen dessert products compared with the control samples. As a result, the ice cream market trend is moving towards a sugar- and fat-free ice cream formulation with excellent texture, structure, and sensory attributes to gain consumer satisfaction.

Ice cream products are an excellent source of dairy and calories. The highest caloric value was observed in the SC1 samples. This increase is due to the increased fat, sucrose, and protein ratios in the SC1 mixture. On the other hand, the lowest caloric value was obtained in SC2, followed by the SC3 and SC4 treatments. The reduction in caloric value was 55.07% for the SC2 treatment, 54.64% for SC3 treatment, and 47.03% for SC4 compared with the SC1 treatment (100%). This result confirms those of Pinto and Dharaiya [41], who detected that replacement of sucrose with sorbitol and bulking fillers resulted in a 20% reduction in calorific value in fat- and sugar-free frozen dessert.

In general, it was found that replacing sugar and fat in ice cream with sweeteners and bulking agents led to a reduction in energy value due to the decrease in the amount added of sweeteners and the decrease in the energy value produced from them. This leads to the production of healthy products for the consumer and nutritional value at the same time.

The changes in microstructure between ice cream with sweeteners and bulk agents or control samples could arise from variances in particle properties (protein and bulking agents). Bulking agents (maltodextrin and polydextrose) impart smoothness and creaminess, improve texture, and structure, and protect against temperature fluctuations to please customers [43]. These data align with those obtained by Agustini et al. [44], who reported that ice cream fortified with spirulina led to a different structure with a fine matrix, obtaining several very small pores. In general, a positive effect of sweeteners and bulking agents would be the enhanced microstructure of ice cream through water binding. It is well known that maltodextrin and polydextrose can play a key role in improving the microstructure, texture, and viscosity of the mixes.

## 5. Conclusions

The effect of sweeteners (stevia, sucralose, and sorbitol) and bulking agents on the chemical, physical, microbiological, and sensory properties and calorific values of ice cream was studied. Four different treatments were studied: the ice cream control (SC1), ice cream with stevia (SC2), ice cream with sucralose (SC3), and ice cream with sorbitol (SC4). The chemical analyses were significantly recorded between all treatments of ice cream. The viscosity, overrun, and hardness values were significantly affected by sweeteners and bulking agents in the ice cream samples. The sensory scores were not significantly affected by sweeteners and bulking agents in the ice cream samples. The reduction in calorific value was 55.07% for the SC2 treatment, 54.64% for the SC3 treatment, and 47.03% for SC4 compared with the SC1 treatment (100%). In SEM micrographs of SC2, SC3, and SC4 treatments, the gel exhibited various structures with a fine network, and it contained a number of very small-sized pores. The current study suggests and concludes that ice cream with sweeteners (stevia, sucralose, and sorbitol) and bulking agents can be promising for marketing. As a result, the ice cream market trend is moving towards a sugar- and fat-free ice cream formulation and with excellent texture, structure, and sensory attributes to gain consumer satisfaction. The replacement of sucrose and fat with sweeteners and bulking agents in the preparation of ice cream can address the issues of current customers who center on normal and healthfully adjusted food sources.

## Figures and Tables

**Figure 1 foods-11-00490-f001:**
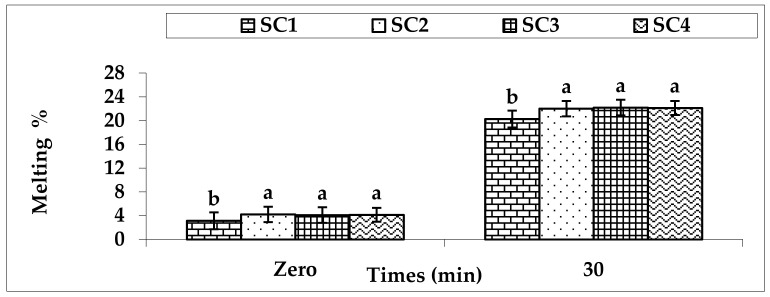
Melting percentage of ice cream treatments. SC1: control without artificial sugar; SC2: ice cream with stevia; SC3: ice cream with sucralose; SC4: ice cream with sorbitol; ^a^ and ^b^: bars within the same time not sharing a common letter are significantly different (*p* < 0.05).

**Figure 2 foods-11-00490-f002:**
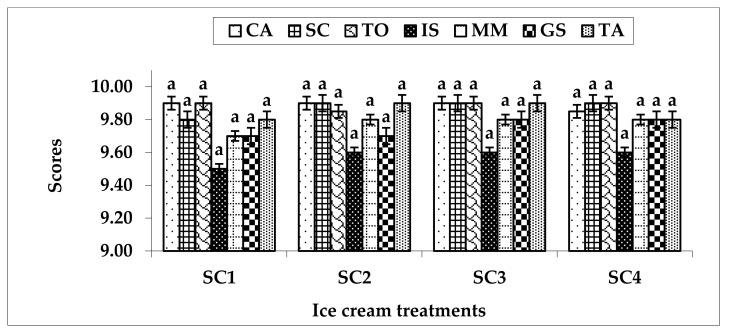
The sensory evaluation of ice cream. CA: color and appearance; SC: structure and consistency; TO: taste and odor; IS: icy structure; MM: melt in mouth; GS: gummy structure; TA: total acceptability, SC1: control without artificial sugar; SC2: ice cream with stevia; SC3: ice cream with sucralose; SC4: ice cream with sorbitol; ^a^: bars within the same characteristic not sharing a common letter are significantly different (*p* < 0.05).

**Figure 3 foods-11-00490-f003:**
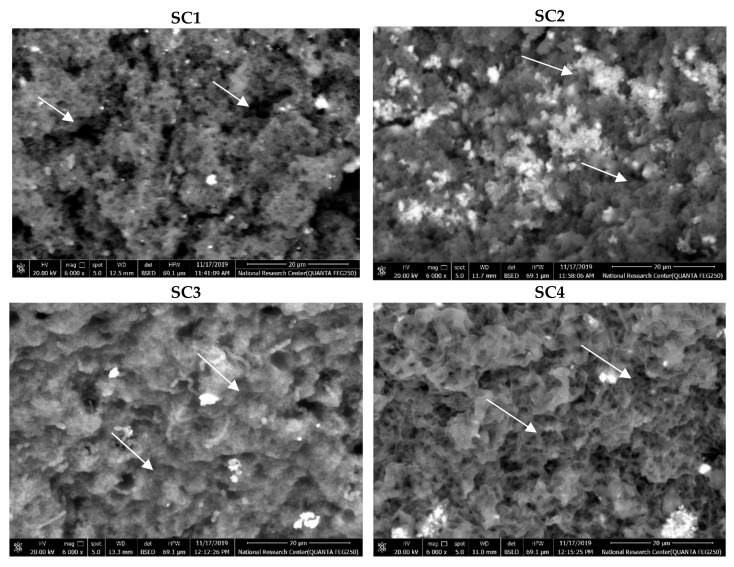
Scanning electron microscopy images of the ice cream treatments. SC1: control without artificial sugar; SC2: ice cream with stevia; SC3: ice cream with sucralose; SC4: ice cream with sorbitol. The arrows indicate the size of pores and the homogeneous structure of samples (or lack thereof).

**Table 1 foods-11-00490-t001:** Formulation of different incorporated ice cream mixes (g/kg)**.**

Ingredients	Control (Ice Cream)	Ice Cream with Stevia	Ice Cream with Sucralose	Ice Cream with Sorbitol
Sugar	150	0	0	0
Stabilizer (CMC)	2.5	2.5	2.5	2.5
Skimmed bovine milk powder	56.5	56.5	56.5	56.5
Fresh skimmed buffalo milk	619.57	790.4	790.7	761
Fresh buffalo cream	171.43	0	0	0
Polydextrose	0	75	75	75
Maltodetrin	0	75	75	75
Stevia	0	0.6	0	0
Sucralose	0	0	0.3	0
Sorbitol	0	0	0	30
Total	1000	1000	1000	1000

CMC: Sodium carboxy methyl cellulose.

**Table 2 foods-11-00490-t002:** Chemical analyses (%, g 100 g^−1^) of ice cream treatments (means ± SE).

Parameters	Ice Cream Treatments			
	SC1	SC2	SC3	SC4
Total solids	31.95 ± 1.54 ^a^	28.01 ± 1.22 ^b^	27.98 ± 1.03 ^b^	28.03 ± 1.01 ^b^
Protein	3.97 ± 0.15 ^a^	3.54 ± 0.20 ^b^	3.60 ± 0.21 ^b^	3.53 ± 0.19 ^b^
Fat	6.08 ± 0.10 ^a^	ND	ND	ND
Ash	1.01 ± 0.00 ^a^	0.86 ± 0.01 ^b^	0.88 ± 0.02 ^b^	0.87 ± 0.01 ^b^
Titratable acidity	0.18 ± 0.01 ^a^	0.17 ± 0.00 ^a^	0.18 ± 0.01 ^a^	0.17 ± 0.01 ^a^

SC1: control without artificial sugar; SC2: ice cream with stevia; SC3: ice cream with sucralose; SC4: ice cream with sorbitol; ND: not detected; ^a^ and ^b^: means with different superscript letters within the same row differ significantly (*p* < 0.05).

**Table 3 foods-11-00490-t003:** Sugar patterns of ice cream treatments with stevia, sucralose, and sorbitol (means ± SE).

Treatments	% (g 100 g^−1^)				
	Glucose	Fructose	Galactose	Sucrose	Lactose
SC1	0.73 ± 0.10 ^a^	0.20 ± 0.07 ^a^	0.30 ± 0.08 ^a^	14.97 ± 1.02 ^a^	3.76 ± 0.19 ^a^
SC2	0.20 ± 0.13 ^b^	ND	0.27 ± 0.04 ^a^	ND	3.60 ± 0.21 ^a^
SC3	0.24 ± 0.16 ^b^	ND	0.25 ± 0.02 ^a^	ND	3.75 ± 0.28 ^a^
SC4	0.27 ± 0.16 ^b^	ND	0.29 ± 0.07 ^a^	ND	3.69 ± 0.20 ^a^

SC1: control without artificial sugar; SC2: ice cream with stevia; SC3: ice cream with sucralose; SC4: ice cream with sorbitol; ND: not detected; ^a^ and ^b^: means with different superscript letters within the same column differ significantly (*p* < 0.05).

**Table 4 foods-11-00490-t004:** Physical analyses of ice cream treatments with stevia, sucralose, and sorbitol (Means ± SE).

Treatments	Overrun %	Viscosity (cP)	Freezing point (°C)	Hardness (N)
SC1	59.82 ± 1.56 ^a^	92.57 ± 4.23 ^a^	−3.35 ± 0.24 ^a^	45.32 ± 3.56 ^a^
SC2	57.99 ± 1.66 ^b^	90.00 ± 3.97 ^b^	−2.29 ± 0.15 ^b^	43.00 ± 4.02 ^b^
SC3	57.95 ± 1.60 ^b^	89.95 ± 3.80 ^b^	−2.31 ± 0.20 ^b^	43.04 ± 4.10 ^b^
SC4	57.97 ± 1.64 ^b^	89.94 ± 4.56 ^b^	−2.32 ± 0.22 ^b^	42.98 ± 3.90 ^b^

SC1: control without artificial sugar; SC2: ice cream with stevia; SC3: ice cream with sucralose; SC4: ice cream with sorbitol; ^a^ and ^b^: means with different superscript letters within the same column differ significantly (*p* < 0.05).

**Table 5 foods-11-00490-t005:** Microbiological analyses (log_10_ CFU g^−1^) of ice cream treatments (means ± SE).

Parameters	Ice Cream Treatments			
	SC1	SC2	SC3	SC4
Aerobic mesophilic bacterial count	4.14 ± 0.41 ^a^	4.30 ± 0.54 ^a^	4.10 ± 0.0.35 ^a^	4.20 ± 0.62 ^a^
Psychrotrophic bacteria	<1	<1	<1	<1

SC1: control without artificial sugar; SC2: ice cream with stevia; SC3: ice cream with sucralose; SC4: ice cream with sorbitol; ^a^: means with different superscript letters within the same row differ significantly (*p* < 0.05).

**Table 6 foods-11-00490-t006:** Total caloric values of the ice cream treatments (means ± SE).

Compounds	Calories	Ice Cream Treatments % (g 100 g^−1^)			
		SC1	SC2	SC3	SC4
Sucrose	3.87	14.97	0	0	0
Lactose	3.87	3.76	3.6	3.75	3.69
Fat	8.79	6.08	0	0	0
Protein	4.27	3.97	3.54	3.6	3.53
Glucose	3.87	0.73	0.2	0.24	0.27
Fructose	3.87	0.20	0	0	0
Galactose	3.87	0.30	0.27	0.25	0.29
Stabilizer (CMC)	3.87	0.25	0.25	0.25	0.25
Polydextrose	1	0	7.5	7.5	7.5
Maltodetrin	3.87	0	7.5	7.5	7.5
Stevia	0	0	0.06	0	0
Sucrolose	3.36	0	0	0.03	0
Sorbitol	3.87	0	0	0	3
Total caloric value (kcal 100 g^−1^)		148.60 ± 2.66 ^a^	68.35 ± 0.97 ^c^	69.37 ± 0.85 ^cd^	80.62 ± 2.75 ^b^

SC1: control without artificial sugar; SC2: ice cream with stevia; SC3: ice cream with sucralose; SC4: ice cream with sorbitol; ^a, b, c^ and ^d^: means with different superscript letters within the same row differ significantly (*p* < 0.05); CMC: Sodium carboxy methyl cellulose.

## Data Availability

The datasets used and/or analyzed during the current study are available from the corresponding author on reasonable request.

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
