# Peer review of "Microstructural, Physicochemical, Microbiological, and Organoleptic Characteristics of Sugar- and Fat-Free Ice Cream from Buffalo Milk"

_foods, 2022, doi:10.3390/foods11030490_

Round 1

Reviewer 1 Report

The research work was properly planned (after doing some preliminary trials to find out the doses of sweetners and fillers), executed and discussed. This type of low caloric food products are need of the market. There are very few corrections of language. The author did not mentioned any side effect of using the sweetners other than sugar. 

Author Response

Top of Form

Open Review

English language and style

( ) Extensive editing of English language and style required 
( ) Moderate English changes required 
(x) English language and style are fine/minor spell check required 
( ) I don't feel qualified to judge about the English language and style 

Yes

Can be improved

Must be improved

Not applicable

Does the introduction provide sufficient background and include all relevant references?

(x)

( )

( )

( )

Is the research design appropriate?

(x)

( )

( )

( )

Are the methods adequately described?

(x)

( )

( )

( )

Are the results clearly presented?

(x)

( )

( )

( )

Are the conclusions supported by the results?

(x)

( )

( )

( )

Comments and Suggestions for Authors

The research work was properly planned (after doing some preliminary trials to find out the doses of sweetners and fillers), executed and discussed. This type of low caloric food products are need of the market. There are very few corrections of language. The author did not mentioned any side effect of using the sweetners other than sugar. 

Submission Date

08 January 2022

Date of this review

20 Jan 2022 07:57:18

Corrections

1- There are very few corrections of language.

  • No comments and suggestions from the reviewer. I reviewed the research.

2- The author did not mentioned any side effect of using the sweetners other than sugar. 

  • Ice cream is frozen dairy products that include healthy and nutritious values. Ice cream is a rich source of sugar, protein, and fat levels. Sucrose level in ice cream products changes between 9% and 30% and fat content ranged from 2% to 15% of the total components. Due to the high prevalence of Type 2 diabetes and obesity among children and adolescents, people are now more aware of their health status and hence conscious of their diet. This health-conscious decision poses a formidable challenge to the ice cream production. Replacement of sucrose and fat with sweeteners and bulk agents in the prepared ice cream can address the issues of current customers who center on normal and healthfully adjusted food sources [34].
  • Ice cream products have a large nutrition value and highly are consumed by different age groups. Due to increasing consumers demands for low energy dairy and foods, dairy products produced with replacement sweeteners and fat have become high popular recently [4]. Other dietary restrictions for the food industry to include consumers with obesity, metabolic syndrome, diabetes, cardiovascular, allergies and those looking for low sugar, fat, and low-calorie products. WHO has determined a world-wide heavy upsurge in diabetic cases, expected to rise by 57.2 million till 2025 in sharp contrast to the diabetic cases of 19.4 million as in 1995 [5,6].

Reviewer 2 Report

This is an interesting study however there are some serious mistakes in manuscript regarding English grammar, therefore professional editing is suggested.

There are also some format problems.

Author Response

Top of Form

Open Review

English language and style

(x) Extensive editing of English language and style required 
( ) Moderate English changes required 
( ) English language and style are fine/minor spell check required 
( ) I don't feel qualified to judge about the English language and style 

Yes

Can be improved

Must be improved

Not applicable

Does the introduction provide sufficient background and include all relevant references?

(x)

( )

( )

( )

Is the research design appropriate?

(x)

( )

( )

( )

Are the methods adequately described?

(x)

( )

( )

( )

Are the results clearly presented?

( )

(x)

( )

( )

Are the conclusions supported by the results?

(x)

( )

( )

( )

Comments and Suggestions for Authors

This is an interesting study however there are some serious mistakes in manuscript regarding English grammar, therefore professional editing is suggested.

There are also some format problems.

Submission Date

08 January 2022

Date of this review

22 Jan 2022 03:43:28

Bottom of Form

© 1996-2022 MDPI (Basel,

 Specific comments

1- Title: Characteristics of microstructural, chemical, physical, microbiological and organoleptical of free sugar fat soft-ice cream durinsg cold storage

the author changes of the title.

  • Microstructural, chemical, physical, microbiological, and organoleptic characteristics of sugar fat free ice cream from buffalo milk

2- lines 19-20-21: The produced soft-ice cream treatments were investigated the effect on microstructural, chemical, physical, microbiological, sensory, and calorific values.

the author reviewed of the sentence.

  • Ice cream treatments were investigated based on microstructural, chemical, physical, microbiological, sensory, and calorific values.

3- lines 27-28: In control soft-ice cream, the highest viscosity, overrun and hardness values (p < 0.05) were detected than the other treatments.

the author changed of the sentence.

  • The highest viscosity, overrun and hardness values (p < 05) were detected in control ice cream than the other treatments.

4- line 31: The highest value of calorific was calculated in SC1 samples (p < 0.05).

the author changed of the sentence.

  • The highest calorific value was calculated in SC1 samples (p < 0.05).

5- lines 31-32-33: On the other hand, the lowest value of calorific was calculated in SC2 followed by SC3, and SC4 treatments.

the author changed of the sentence.

  • On the other hand, the lowest calorific value was calculated in SC2 followed by SC3, and SC4 treatments.

6- lines 34-35: it's contained a cohesive structure and a small number of pores in size within the structure.

 the author changed of the sentence.

  • it's contained a cohesive structure with small pores in size within the structure.

7- lines 39-40: Ice cream products are a rich source of fat carbohydrate and protein which reflects towards its energy value.

the author changed of the sentence.

  • Ice creams are a rich source of fat, carbohydrate, and protein which reflects towards its energy value.

8- line 41: The consumers' awareness for functional and healthier

 the author changed of the sentence.

  • The consumers' awareness of functional and healthier

9- line 41-42-43: Levels of sucrose in ice cream were ranged from 9 to 28% of the total ingredients. Fat levels of ice cream changed between 3 and 15% of the total materials [3].

  the author changed of the sentence.

  • Sucrose levels in ice cream are ranged from 9% to 28% and fat levels between 3% and 15% of the total ingredients [3].

10- lines 45-46: Ice cream products a large nutrition value and highly consumed by different age groups.

   the author changed of the sentence.

  • Ice cream products have a large nutrition value and highly are consumed by different age groups.

11- line 47: …. produced with replacement sweetener and fat have becomes high …

   the author changed of the sentence.

  • … produced with replacement sweeteners and fat have become high …

12- line 48: … for the food industry to consider include consumers ….

   the author changed of the sentence.

  • … for the food industry to include consumers …

13- line 51: …. heavy upsurge in the diabetic cases …..

 the author changed of the sentence.

  • … heavy upsurge in diabetic cases …

14- line 54: they are prefect for those who are looking for low sugar, ….

   the author changed of the sentence.

  • they are preferred for those looking for low sugar

15- line 58: .. and later by adding freezing point depressant like sorbitol ..

  the author changed of the sentence.

  • … and later by adding a freezing point depressant like sorbitol …

16- line 65:  .. pure taste to ice cream; thus has a crucial ….

the author changed of the sentence.

  • … pure taste to ice cream and thus has a crucial ….

17- line 66: In order to produce ice cream which is sufficiently …..

the author changed of the sentence.

  • To produce ice cream which is sufficiently …..

18- lines 68-69-70-71: When fat and sucrose, which is used generally at the rate of about 6 and 15% respectively, is to be replaced by intense sweetener, the requirement is in milligram or ppm only.

the author changed of the sentence.

  • When fat levels are used about 3% to 15% and sucrose levels between 9% and 28% in ice cream and the sugar and fat are to be replaced by intense sweeteners and bulk agents.

19- line 74: However, addition of bulk agents …..

the author changed of the sentence.

  • However, the addition of bulk agents …..

20- lines 76-77-78: Bulking agents impart creaminess, smoothness, improve texture and provide a mouth feel and protection against temperature fluctuation to please customers [10].

the author changed of the sentence.

  • Bulking agents impart creaminess and smoothness improves texture and provides a mouthfeel and protection against temperature fluctuation to please customers [10].

21- lines 79-80: Stevia powder (Stevia rebaudiana) is characterized as a natural sweetener

the author changed of the sentence.

  • Stevia powder (Stevia rebaudiana) is a natural sweetener

22- line 82: it a good alternative of sugar for

the author changed of the sentence.

  • it a good alternative to sugar for …

23- lines 83-84: Stevia is recognized as safe supplements by JECFA ….

the author changed of the sentence.

  • Stevia is recognized as safe supplement by JECFA …

24- lines 87-88-89: … it a good alternative of sugar [16]. Sorbitol containing sugar-free products contain low glycemic index (GI) and it is about 60% sweet than sucrose ….

the author changed of the sentence.

  • …. it a good alternative to sugar [16]. Sorbitol containing sugar-free products contains a low glycemic index (GI) and it is about 60% more sweetly than sucrose ….

25- line 90: ………. which result in ….

the author changed of the sentence.

  • …. which results in ….

26- line 93: ………. that include the healthy and nutritious ………………

the author changed of the sentence.

  • ……. that include healthy and nutritious …………….

27- line 95: fat content ranged from 2 to 15% of ………….

the author changed of the sentence.

  • ….. fat content ranges from 2% to 15% of …………

28- lines 98-99: ……… poses a formidable challenge to the ice cream production.

the author changed of the sentence.

  • ….. poses a formidable challenge to ice cream production.

29- lines 99-100-101: a free sugar fat soft-ice cream formulation and with excellent texture, structure and sensory attributes to gain the consumers satisfaction.

the author changed of the sentence.

  • ………. a sugar fat free ice cream formulation with excellent texture, structure, and sensory attributes to gain the consumer satisfaction.

30- lines 128-129: Different mixes were whipped during freezing in horizontal batch freezer ……

the author changed of the sentence.

  • Different mixes were whipped during freezing in a horizontal batch freezer ………….

31- lines 130-31: ……… for 24 hours in deep freezer and kept at -20ºC.

the author changed of the sentence.

  • …………. for 24 hours in a deep freezer and kept at -20ºC.

32- lines 134 to 142: Initial experiments were conducted in the making of soft-ice cream using different amounts of stevia, sorbitol, and sucralose. It was observed that the best value of stevia (0.06%), sucralose (0.03%), and sorbitol (3%) in the soft-ice cream mixes with performing some tests such as sensory evaluation and physical tests. Also, initial tests were conducted in the preparation of soft-ice cream using different values of maltodextrin and polydextrose as bulking agents impart improve texture, smoothness, and protection against temperature fluctuation to please customers. The best value of maltodextrin was 7.5% and polydextrose was 7.5% in the soft-ice cream production (Results not shown).

the author changed of the sentence.

  • Initial experiments were conducted to make ice cream using different amounts of stevia, sorbitol, and sucralose. It was observed that the best value of stevia (0.06%), sucralose (0.03%), and sorbitol (3%) in the ice cream mixes with performing some tests such as sensory evaluation and physical tests. Also, initial tests were conducted to prepare of ice cream using different values of maltodextrin and polydextrose as bulking agents. The best value of maltodextrin was 7.5% and polydextrose was 7.5% in the ice cream production (Results not shown).

33- line 144: …… determined by drying 2 to 3 grams at 105ºC using ……….

the author changed of the sentence.

  • …….. determined by drying 2 to 3 g at 105ºC using …..

34- line 147: equation

TP =

TN  *  6.38 

……………………. (1)

the author changed of the equation.

  • TP = TN * 6.38 …………………………………….……………….. (1)
  • where: TP is the total protein, TN: total nitrogen.

35- line 162: 20 µL of extract was injected in the …………….

the author changed of the sentence.

  • 20 µL of the extract was injected in the ………….

36- line 168: Colifrom bacterial were observed ………………

the author changed of the sentence.

  • Coliform bacterial were observed ……………………

37- lines 173 - 177: Freezing point was calculated according to [28]. The viscosity (centi Poise) of mixes were detected according to Brookfield measurement at 5ºC with a spindle No. #07 for 50 rpm and the reading recorded after 30 seconds in 250 ml cup of ice cream mixes. Overrun of ice-cream samples was calculated by using the method given by [29] as follows:

Overrun  =

A - B/  B × 100

………………………….. (2)

the author changed of the sentence.

  • The freezing point was calculated according to [28] applying the digital thermometer. The viscosity (cP) of mixes was detected according to Brookfield measurement at 5ºC with a spindle No. #07 for 50 rpm and the reading recorded after 30 s in 250 ml cup of ice cream mixes. Overrun of ice cream samples was calculated by using the method given by [29] as follows:

 the author changed of the equation.

  • Overrun = [(A – B) / B] * 100 ……………………..…………………… (2)

38- lines 178-179: where: A is the weight of volume of mix; B is the weight of the same volume of ice-cream.

the author changed of the sentence.

  • where: A is the weight of a volume of the mix and B is the weight of the same volume of ice cream.

39- line 180 - 182: Melting resistance of ice cream was measured according to [30]. 100 grams of ice cream sample was placed into wire mesh (6 pores /cm2) over a glass funnel fitted on conical flask at ambient temperature.

the author changed of the sentence.

  • The melting resistance of ice cream was measured according to [30]. 100 g of ice cream sample was placed into wire mesh (6 pores /cm2) over a glass funnel fitted on a conical flask at ambient temperature.

40- lines 184 - 190: Hardness of ice cream samples was determined by a Universal Testing Machine (TMS-Pro, Japan) equipped with (250 lbf) load cell and connected to a computer programmed with Texture ProTM texture analysis software (Texture ProTM, program, DEVTPA with the hold). A flat rod probe (49.95 mm in diameter) was used to uniaxial compress the ice cream samples to 50% of their original height. The hardness was adjusted to a test speed 60 mm/s; trigger force 1 N, deformation 25% and holding time 2 seconds between cycles at -20ºC.

the author changed of the sentence.

  • The hardness of ice cream samples was determined by a Universal Testing Machine (TMS-Pro, Japan) equipped with a load cell (250 Ibf) and connected to a computer programmed with Texture ProTM texture analysis software (Texture ProTM, program, DEVTPA with the hold). A flat rod probe (49.95 mm in diameter) was used to uniaxial compress the ice cream samples to 50% of their original height. The hardness was adjusted to a test speed of 60 mm/s; trigger force 1 N, deformation 25% and holding time 2 s between cycles at -20ºC.

41- lines 193 - 196: ……. the Dairy Science Department, Faulty of Agriculture, Benha University Egypt, was assembled. Panel members were selected based on their interest in the sensory evaluations of ice cream and were trained by testing commercial ice cream. Samples (50 grams) were put into a group to the 15 test panelists.

the author changed of the sentence.

  • …….. the Dairy Science Department, Faculty of Agriculture, Benha University Egypt, was assembled. Panel members were selected based on their interest in the sensory evaluations of ice cream and were trained by testing commercial ice cream. Samples (50 g) were put into a group of 15 test panelists.

42- line 204: equation

Calorific value = % sugar x 3.87 + % fat x 8.79 + % protein x 4.27

…….…… (3)

the author changed of the equation.

  • Calorific value = % sugar * 3.87 + % fat * 8.79 + protein * 4.27 …………….. (3)

43- lines 208 - 211: ………… at 6000 magnifications were recorded using a SEM (FEI company, Netherlands) model quanta 250 FEG (field emission gun) attached with EDX unit (energy dispersive x-ray analyses), The images were taken at an excitation voltage of 20 KV, at different …………………..

the author changed of the sentence.

  • ……….. at 6000 magnifications were recorded using SEM (FEI Company, Netherlands) model quanta 250 FEG (field emission gun) attached with EDX unit (energy dispersive x-ray analyses). The images were taken at an excitation voltage of 20 kV, at different …………..

44- line 214: Results of all treatments during storage periods were ………………

the author changed of the sentence.

  • Results were subjected …………………

45- line 217: Applied static model is as follows:

the author changed of the sentence.

  • The applied static model is as follows:

46- line 218: equation

Yij = µ + Ti+ eij

……………………………………………………..…… (4)

the author changed of the equation.

  • Yij = µ + Ti+ eij ……………………………………….………….………… (4)

47- lines 224 - 227: …………… affected by addition of sweeteners and bulk agents in soft-ice cream. Addition of sweeteners (stevia, sucralose, and sorbitol) and bulk agents had significant (p < 0.05) effect on the change of total solids, fat, protein, and ash values of all treatments.

the author changed of the sentence.

  • ………… affected by the addition of sweeteners and bulk agents in ice cream. The addition of sweeteners (stevia, sucralose, and sorbitol) and bulk agents had a significant (p <05) effect on the change of total solids, fat, protein, and ash values of all treatments.

48- lines 229-230: Fat values were recorded 6.08±0.10% (w/w) in SC1 samples and in the other treatments were not detected.

the author changed of the sentence.

  • Fat value was 6.08±0.10% (w/w) in SC1 sample and was not detected in the other treatments.

49- lines 232 - 235: The highest levels of total solid, fat, protein, and ash were detected in control samples (SC1). Replacement of sucrose with sweeteners and bulking agents had effect the chemical properties of the soft-ice cream mixture.

the author changed of the sentence.

  • The highest levels of total solids, fat, protein, and ash were detected in control samples (SC1). Replacement of sucrose with sweeteners and bulking agents influenced the chemical properties of the ice cream

50- lines 239 - 241: There is no significant difference (p > 0.05) between any two means, within the same row have the same superscript letter.

the author changed of the sentence.

  • There is no significant difference (p > 0.05) between any two means, within the same row has the same superscript letter.

51- line 244: …………… bulking agents had significant (p < 0.05) effect ….

the author changed of the sentence.

  • ………. bulking agents had a significant (p <05) effect ………….

52- lines 246-247: … in SC1 treatment compared the other treatments.

the author changed of the sentence.

  • …… in SC1 treatment compared to the other treatments.

53- line 250: Addition of sweeteners and bulking ……………….

the author changed of the sentence.

  • The addition of sweeteners and bulking ………………

54- line 255: Sugar patterns of soft-ice cream with stevia, sucrolose and sorbitol (Means ± SE)

the author changed of the sentence.

  • Sugar patterns of ice cream with stevia, sucralose, and sorbitol (Means ± SE).

55- lines 256 - 259: ………….. SC3: soft-ice cream with sucrolose; SC4: soft-ice cream with sorbitol; ND: not detected; a and b: There is no significant difference (p > 0.05) between any two means, within the same column have the same superscript letter.

the author changed of the sentence.

  • ……………. SC3: ice cream with sucralose; SC4: ice cream with sorbitol; ND: not detected; a and b: There is no significant difference (p > 0.05) between any two means, within the same column has the same superscript letter.

56- line 262: ……….. it directly has relation ……………

the author changed of the sentence.

  • …….. it directly has a relation ………….

57- line 265 - 267: Overrun of ice cream samples are shows in Table 4. Overrun was significantly affected (p < 0.05) with replacement of sucrose with sweeteners and ……..

the author changed of the sentence.

  • Overrun of ice cream samples are shown in Table 4. Overrun was significantly affected (p <05) with sucrose replacement using sweeteners and …………….

58- line 273 - 275: There is no significant difference (p > 0.05) between any two means, within the same column have the same superscript letter.

the author changed of the sentence.

  • There is no significant difference (p > 0.05) between any two means, within the same column has the same superscript letter.

59- line 277: Viscosity had been considered …………………

the author changed of the sentence.

  • The viscosity had been considered ……………….

60- line 292: Hardness of the products at …………

the author changed of the sentence.

  • The hardness of the products at ………….

61- lines 296-297: Melting and freezing point decreased as the level of water-soluble ingredients increase.

the author changed of the sentence.

  • Melting and freezing points decreased as the level of water-soluble ingredients increased.

62- lines 300-301: The results demonstrated that there were significant differences ……

the author changed of the sentence.

  • The results demonstrated significant differences ………………

63- lines 304-306: Melting resistance is expressed as the loss of weight tested sample compared to its initial weight. Additionally, melting resistance is a good indicator for the structure of product.  

the author changed of the sentence.

  • The melting % is expressed as the loss of weight tested sample compared to its initial weight. Additionally, melting % is a good indicator for the structure of the product.

64- lines 322-323: The count of total psychrotrophic bacteria in all treatments were <1 log10 CFU g-1 log10 CFU g-1.

the author changed of the sentence.

  • The count of total psychrotrophic bacteria in all treatments was <1 log10 CFU g-1.

65- lines 327-329: There is no significant difference (p > 0.05) between any two means, within the same row have the same superscript letter.

the author changed of the sentence.

  • There is no significant difference (p > 0.05) between any two means, within the same row has the same superscript letter.

66- line 333: There were not significant differences ……………

the author changed of the sentence.

  • There were no significant differences …………

67- line 334: Structure and consistency …………

the author changed of the sentence.

  • The structure and consistency ……………

68- line: Melt in mouth scores varied …………

the author changed of the sentence.

  • The melt in mouth scores varied ……………

69- line 346: The caloric value of soft-ice cream was computerized by taking …………..

the author changed of the sentence.

  • The caloric value of ice cream was calculated by taking ………………

70- line 350: The lowest value of caloric was …………………

the author changed of the sentence.

  • The lowest caloric value was ………………….

71- line 352: The highest value of caloric was observed …………….

the author changed of the sentence.

  • The highest caloric value was observed ………….

72- lines 356-358: There is no significant difference (p > 0.05) between any two means, within the same row have the same superscript letter.

the author changed of the sentence.

  • There is no significant difference (p > 05) between any two means, within the same row has the same superscript letter.

73- lines 363-364: …. it's contained a cohesive structure and a small number of pores in size within the structure.  

the author changed of the sentence.

  • .. it's contained a cohesive structure with the small pores in size within the structure.

74- lines 367-370: …….. and the gel appeared heterogeneity in the pores size. Addition of maltodextrin and polydextrose led to a homogeneous microstructure with a fine matrix, obtaining numbers very small pores.

the author changed of the sentence.

  • ….. and the gel appeared heterogeneity in the size of pores. The addition of maltodextrin and polydextrose led to a homogeneous microstructure with a fine matrix, obtaining very small pores.

75- lines 371-372: ……… matrix in frozen yogurt with maltodextrin and polydextrose.

the author changed of the sentence.

  • ………….. matrix in ice cream with maltodextrin and polydextrose.

76- line 378: The main aim of the current study was evaluated a novel ……………

the author changed of the sentence.

  • The main aim of the current study was to evaluate a novel ……..

77- lines 392-395: Consequently, several studies had been determined in developing new functional ice creams with ingredients such as dietary fibers [1], low glycemic index (GI) sweeteners [18] and alternative sweeteners [2].

the author changed of the sentence.

  • I deleted it.

78- lines 397-402: With increased consumer attentiveness for efficient and improved dairy and foods various new technologies has come to the fore for production of such products. Ice cream is one of the most served and loved frozen dairy products but is high in levels of sugar (15%) and fat (6%) therefore; formulating its sugar and fat free version will serve in good cause for decreasing the extra-calorie intake and make it healthier.

the author changed of the sentence.

  • With increased consumer attentiveness for efficient and improved dairy and foods, various new technologies have come to the fore to produce such products. Ice cream is one of the most served and loved frozen dairy products but is high in sugar levels (15%) and fat (6%) therefore; formulating its sugar and fat free version will serve in a good cause for decreasing the extra-calorie intake and make it healthier.

79- lines 404-406: Low GI dairy are important in dietary management as they allow slow movement of glucose into the blood resulting in very low rise in blood glucose, obesity, and insulin percent's [35].

the author changed of the sentence.

  • Low sugar dairy is important in dietary management as they allow slow movement of glucose into the blood resulting in a very low rise in blood glucose, obesity, and insulin levels [35].

80- lines 412-413: Similar data were recorded by [37] who reported significant effect ……

the author changed of the sentence.

  • Similar data were recorded by [37] who reported a significant effect …….

81- line 428-429: …. several healthy trends in the dairy products in recent years.

the author changed of the sentence.

  • …… several healthy trends in dairy products in recent years.

82- lines 435-436: ….. the viscosity value of treatments containing sugar alcohols and HFCS were lower than that of control samples.

the author changed of the sentence.

  • …. the viscosity value of treatments containing sugar alcohols and HFCS was lower than that of control samples.

83- lines 441-442: Also, [39] reported that the overrun value of treatments containing sugar alcohols and HFCS were lower than that ……….

the author changed of the sentence.

  • Also, [39] reported that the overrun value of treatments containing sugar alcohols and HFCS was lower than that …………..

84- line 443: Levels of freezing point were affected with replacement ………………

the author changed of the sentence.

  • The freezing point values were affected with the replacement …………..

85- lines 450-451: The highest value of melting was detected in …………….

the author changed of the sentence.

  • The highest melting % was detected in …………………

86- line 455: lower molecular have a decreased melting resistance as compared

the author changed of the sentence.

  • lower molecular have a decreased melting % compared ………………

87- lines 463-464: ………….. on counts of standard plate in yogurt ice cream ……………

the author changed of the sentence.

  • ……. on counts of standard plate in ice cream ………….

88- lines 472-473: The scores of taste & odor, structure & consistency, and total acceptability total acceptability were not affected by replacement …….

the author changed of the sentence.

  • The scores of taste & odor, structure & consistency, and total acceptability total acceptability were not affected by the replacement ……

89- lines 474-475: ………….. who reported an improving in the sensory scores with addition of maltodextrin amount …………………..

the author changed of the sentence.

  • ………… who reported an improvement in the sensory scores with the addition of maltodextrin ………………..

90- line 477: …….. sensory scores was relatively higher in replacement of …………..

the author changed of the sentence.

  • …… sensory scores was relatively higher in the replacement of …………..

91- line 478: ….. who detected highest increase of ……………

the author changed of the sentence.

  • ………….. who detected the highest increase of ………..

92- lines 480-481: …. towards a free sugar fat soft-ice cream formulation and with excellent ………….

the author changed of the sentence.

  • ……………. towards a sugar fat free ice cream formulation with excellent ……….

93- lines 483-486: Ice cream products are an excellent source of dairy and foods calorie. The fact that the components of ice cream are about completely assimilated produces ice cream and particularly desirable food for growth children and for human who need to put on weight.

the author changed of the sentence.

  • Ice cream products are an excellent source of dairy and foods calories. The fact that the components of ice cream are about completely assimilated produces ice cream.

94- lines 493-494: …… bulk fillers resulted in 20% reduction in calorific value in the fat sugar …..

the author changed of the sentence.

  • ….. bulk fillers resulted in a 20% reduction in calorific value in the fat sugar …….

95- lines 495-496: ……………… it was found that the process of replacing sugar and fat by sweeteners and bulking agents in the soft-ice cream ………………..

the author changed of the sentence.

  • ……. it was found that replacing sugar and fat with sweeteners and bulking agents in the ice cream …………..

96- lines 505-506: …………….. spirulina led to a various structure with a fine …………….

the author changed of the sentence.

  • ……….. spirulina led to a different structure with a fine ………….

97- lines 506-510: In general, a positive effect of sweeteners and bulking agents would be to enhance the microstructure of ice cream though the water binding. It is well known that maltodextrin and polydextrose can play a key role in improving microstructure, texture, and viscosity of the mixes.

the author changed of the sentence.

  • In general, a positive effect of sweeteners and bulking agents would be to enhance the microstructure of ice cream through the water binding. It is well known that maltodextrin and polydextrose can play a key role in improving the microstructure, texture, and viscosity of the mixes.

98- line 512: Effect of sweeteners (stevia, sucralose, and sorbitol) …………

the author changed of the sentence.

  • The effect of sweeteners (stevia, sucralose, and sorbitol) ……………

99- line 516: Chemical analyses were significantly ……………….

the author changed of the sentence.

  • The chemical analyses were significantly …………………….

100- line 517: Viscosity, overrun, and hardness values …………………

the author changed of the sentence.

  • The viscosity, overrun, and hardness values …………………

101- lines 522-523: …………. SC2, SC3 and SC4 treatments, the gel exhibited a various structure with a fine …………….

the author changed of the sentence.

  • …………… SC2, SC3 and SC4 treatments, the gel exhibited various structures with a fine ………………..

102- lines 523-524: ………….. it's contained numbers very small pores in size.

the author changed of the sentence.

  • ……………. it's contained numbers of very small pores in size.

Author Response

Open Review

English language and style

(x) Extensive editing of English language and style required 
( ) Moderate English changes required 
( ) English language and style are fine/minor spell check required 
( ) I don't feel qualified to judge about the English language and style 

Yes

Can be improved

Must be improved

Not applicable

Does the introduction provide sufficient background and include all relevant references?

( )

(x)

( )

( )

Is the research design appropriate?

(x)

( )

( )

( )

Are the methods adequately described?

( )

(x)

( )

( )

Are the results clearly presented?

( )

(x)

( )

( )

Are the conclusions supported by the results?

( )

(x)

( )

( )

Comments and Suggestions for Authors

Submission Date

08 January 2022

Date of this review

21 Jan 2022 17:01:17

 General comments

1- English language of the manuscript needs correction by an English native speaker

Au:

- the manuscript reviewed by an English native speaker

2- There are many typographical mistakes

Au:

-  Typographical mistakes in manuscript are corrected.

3- In some cases, wrong terminology or words and phrases with wrong scientific meaning are used. Therefore, the manuscript needs correction by a senior researcher on this topic.

Au:

- the manuscript reviewed by a senior researcher.

4 - Title

Why the term soft ice cream is used? Soft-frozen products are those that are consumed directly and immediately after dynamic freezing with no hardening step. Unless you mean soft-scooped ice cream, but there is not a justified reason or values of regular hardness in this study to call your products soft-scooped.

Title must be as: Microstructural, chemical, physical, microbiological and organoleptic characteristics of sugar fat free ice cream (from buffalo milk?)

Au:

- the author changes of the title.

  • Microstructural, chemical, physical, microbiological, and organoleptic characteristics of sugar fat free ice cream from buffalo milk

5- Abstract and Introduction need rewriting. The relative corrections below are indicatives.

Au:

-  Abstract and Introduction are rewrite.

6- Discussion is general as to be introduction. For example, there is discussion about GI without having analyzing it. There in not substantial comparison with results from similar studies.

Au:

-  the author deleted of the GI from the manuscript.

Specific comments

Introduction

7- Lines 40-41: Reference is not used in the right way. Hence, the sentence ‘Levels of sucrose in ice cream were ranged 40 from 9 to 28% of the total ingredients. Fat levels of ice cream changed between 3 and 15% of the total materials [3]’ should be written in the present tense as follows: Levels of sucrose in ice cream are ranged from 9% to 28% and fat levels between 3% and 15% of the total ingredients [3].

Au:

- Sucrose levels in ice cream are ranged from 9% to 28% and fat levels between 3% and 15% of the total ingredients [3].

8- Line 50: they are prefect for…. Please correct to they are preferred for …

Au:

- …………… they are preferred for those looking ……………..

9- Line 53: …loss in freezing point….Probably you mean … in freezing point depress.

Au:

-  ……………….. loss in freezing point depress.

10- Line 67: …milligram per 100 g or per 1kg?

Au:

- …milligram per 100 g ………………….

11- Lines 69-70: However, addition of bulk agents will give in somewhat hard products because they lower the freezing point much lower than sucrose. This is not true. See your results in Table 4.

Au:

- However, the addition of bulk agents will give in somewhat smooth products because they lower the freezing point much lower than sucrose. Bulking agents impart creaminess and smoothness improves texture and provides a mouthfeel and protection against temperature fluctuation to please customers [10].

12- Line 77: zero calories)… per gram?

Au:

- one gram gives zero calories

13- Line 90: …about 600 times that of sucrose…. 600 times of what?

Au:

- Sucralose is an artificial sweetener and is about 600 times sweeter than sucrose.

Materials and Methods

14- Line 101: Milk was also from buffalos? If so, then it is better to state it in the title, i.e. buffalo ice cream

Au:

- yes, I added in the title.

-" Microstructural, chemical, physical, microbiological, and organoleptic characteristics of sugar fat free ice cream from buffalo milk"

15- Line 103: Skimmed milk powder was bovine?

Au:

- yes, I added in the materials.

" Skimmed bovine milk powder"

Table 1

16- Title: Preparation Formulation of different…

Au:

- the author changes of the table title.

" Table 1. Formulation of different incorporated ice cream mixes (g/ kg).

17- The total weights of control and ice cream with stevia are not 1000 (they are 977.5 and 999.4 respectively)

Au:

- the author reviewed and corrected of the total weights in table 1. This is shown in the table.

18- Line 138-139: Total protein (TP) levels of all samples were calculated…..

Au: the author reviewed and corrected

- Total protein (TP) levels of all samples were calculated by………..

19- Lines 151-153: Which was the column?

Au:

- DCou-14A, Guard column Sc-LcShodex

20- Line 157: … total aerobic mesophilic bacterial count…

Au: the author corrects these changes in the manuscript.

- The total aerobic mesophilic bacterial count was carried out in plate ……………

21- Line 164: Freezing point calculation needs more information.

Au: the author adds more information for freezing point.

- The freezing point was determined according to [28] applying the digital thermometer.

22- Overrun = [(A – B)/ B] × 100

Au: the author corrects these changes in the equation.

- Overrun = [(A – B)/ B] × 100

23- Line 195: Samples were prepared as described by [32]. Do you mean that you dried the ice cream? This analysis needs more information.

Au: the author adds more information for preparation of samples as follows:

- as follows: The samples were fixed on an iron stub and then made electrically conductive by coating it (in a vacuum chamber) with a thin layer of gold for 40 s. The moisture of samples was entirely removed by placing the samples in an air-tight desiccator containing silica gel. The weight of samples was periodically determined until constant weight to confirm the complete removal of moisture.

Results

24- Line 268: The freezing point values of soft-ice cream mix samples…

Au: the author corrects these changes in the manuscript.

- The freezing point values of ice cream mix samples ……………

25- Line 291: The lower values of melted ice cream melting resistance….

Au: the author corrects these changes in the manuscript.

- The lower values of melted ice cream melting resistance…. ……………

26- Line 348: … in frozen yogurt (????) with maltodextrin and polydextrose.

Au: the author corrects these typographical mistakes in the manuscript.

- ……….. in ice cream with maltodextrin and polydextrose.

 27- Figure 3. What do the arrows indicate? Explain in the text or in the figure caption.

Au: the author adds in the figure.

-  the arrows indicate to the size of pores and homogeneous structure or not of samples.

28- Table 6. Why the values of some sugars, e.g. glucose and lactose are different from those presented in Table 3?

Au: the author corrects these typographical mistakes in the table 6 according to table 3.

Discussion

29- Lines 389-390: .. levels of fat were not detected in SC2, SC3 and SC4 treatments. This was expected since you did not add fat at all in these formulations.

Au: the author deletes this sentence from the manuscript.

30- Line 425: … which resulted in increase decrease of the….

Au:

- ……………… which resulted in increase of the required time for melting %.

31- Line 426: … Also, this increase in melting is due… (There is a confusion between melting (%) and melting resistance measured at a certain time i.e., at the 30 min in this study. These two parameters are different each other. The sample with the higher melting (%) at a certain time is that with the lower melting resistance).

Au: the author corrects these typographical mistakes

- Also, this increase is due to the decrease of the total solids, fat, sucrose, ash, and protein ratios in the mixtures. Also, they reported that sugars with lower molecular have a decreased melting % compared to those with higher molecular weight. Many factors can affect the melting % of ice cream samples: ice crystal size, fat destabilization and the consistency coefficient of the mixture [40]. These data are confirmed by [41] who found the frozen dessert melting % showed a decrease in the values of melting % was detected for SC1 treatment.

32- Line 432: …. values of melting rate was detected for SC1… You did not measure melting rate. You measured the melting (% of initial amount) at 30 min. This is different.

Au: the author corrects these typographical mistakes

-this test determines the melting % in the manuscript and it corrects this in the text.

References

33- 10. ….2018b. Why b?

Au: there two references for the same author as follows:

- 3. Goff, H.D. Ice cream and frozen desserts: product types, in Smithers, G.W. (Ed), Reference Module in Food Science, Elsevier, Oxford, 2018a, pp. 1-6.

- 10. Goff, H.D. The structure and properties of ice cream and frozen desserts, in Shahidi, F.; Varelis, P. (Eds), Encyclopedia of food Chemistry, Vol. 3, 1st ed., Guelph, 2018b.

Round 2

Reviewer 2 Report

The revised version is ok.

Author Response

Open Review

English language and style

( ) Extensive editing of English language and style required 
( ) Moderate English changes required 
(x) English language and style are fine/minor spell check required 
( ) I don't feel qualified to judge about the English language and style 

Yes

Can be improved

Must be improved

Not applicable

Does the introduction provide sufficient background and include all relevant references?

(x)

( )

( )

( )

Is the research design appropriate?

(x)

( )

( )

( )

Are the methods adequately described?

(x)

( )

( )

( )

Are the results clearly presented?

(x)

( )

( )

( )

Are the conclusions supported by the results?

(x)

( )

( )

( )

Comments and Suggestions for Authors

The revised version is ok.

Submission Date

08 January 2022

Date of this review

25 Jan 2022 23:47:07

Bottom of Form

© 1996-2022 MDPI (Basel, Switzerland) unless otherwise stated

Comments and Suggestions for Authors

The revised version is ok.

Au:

Top of Form

Bottom of Form

No comments and suggestions from the reviewer.

Reviewer 3 Report

Reviewer’s comments to authors

Title

Please change the title to ‘Microstructural, physicochemical, microbiological, and organoleptic characteristics of sugar fat free ice cream from buffalo milk’

Introduction

Correct lines 113-115 as follows:

Reduction of sucrose or fat to produce sucrose and fat free ice cream products affects the solids and the freezing point depress. The first can be compensated by bulking agent like maltodextrin and polydextrose, and the later by adding a freezing point depressant like sorbitol

The same for lines 143-144.

Sorbitol containing sugar-free products have a lower glycemic index and are about 60% more sweet than those containing sucrose (one gram gives 2.60 calories)’

The same for line 148:

Ice cream is frozen dairy product that includes healthy and nutritious aspects.

Materials and Methods

Line 164 (now 332): The information about freezing point calculation is not sufficient. Because you used thermometer to measure this parameter, it seems that you mean the temperature of ice cream when exited the ice cream machine. Hence, analytical description of this analysis is necessary.

Line 340: The weight of melted ice cream expressed as % of initial weight of ice cream was measured according …

Results

The footnotes under each Table should be corrected accordingly as follows: Means with different superscript letters within the same row (or the same column) differ significantly (p <0.05).  

Author Response

Open Review

English language and style

( ) Extensive editing of English language and style required 
(x) Moderate English changes required 
( ) English language and style are fine/minor spell check required 
( ) I don't feel qualified to judge about the English language and style 

Yes

Can be improved

Must be improved

Not applicable

Does the introduction provide sufficient background and include all relevant references?

(x)

( )

( )

( )

Is the research design appropriate?

(x)

( )

( )

( )

Are the methods adequately described?

( )

(x)

( )

( )

Are the results clearly presented?

(x)

( )

( )

( )

Are the conclusions supported by the results?

(x)

( )

( )

( )

Comments and Suggestions for Authors

Reviewer’s comments to authors

Title

Please change the title to ‘Microstructural, physicochemical, microbiological, and organoleptic characteristics of sugar fat free ice cream from buffalo milk’

Introduction

Correct lines 113-115 as follows:

Reduction of sucrose or fat to produce sucrose and fat free ice cream products affects the solids and the freezing point depress. The first can be compensated by bulking agent like maltodextrin and polydextrose, and the later by adding a freezing point depressant like sorbitol

The same for lines 143-144.

Sorbitol containing sugar-free products have a lower glycemic index and are about 60% more sweet than those containing sucrose (one gram gives 2.60 calories)’

The same for line 148:

Ice cream is frozen dairy product that includes healthy and nutritious aspects.

Materials and Methods

Line 164 (now 332): The information about freezing point calculation is not sufficient. Because you used thermometer to measure this parameter, it seems that you mean the temperature of ice cream when exited the ice cream machine. Hence, analytical description of this analysis is necessary.

Line 340: The weight of melted ice cream expressed as % of initial weight of ice cream was measured according …

Results

The footnotes under each Table should be corrected accordingly as follows: Means with different superscript letters within the same row (or the same column) differ significantly (p <0.05).  

Submission Date

08 January 2022

Date of this review

27 Jan 2022 18:08:31

Bottom of Form

© 1996-2022 MDPI (Basel, Switzerland) unless otherwise stated

Corrections of comments

Title

Please change the title to ‘Microstructural, physicochemical, microbiological, and organoleptic characteristics of sugar fat free ice cream from buffalo milk’

Au:

- The author changes of the title.

- Microstructural, physicochemical, microbiological, and organoleptic characteristics of sugar fat free ice cream from buffalo milk

Introduction

Correct lines 113-115 as follows:

Reduction of sucrose or fat to produce sucrose and fat free ice cream products affects the solids and the freezing point depress. The first can be compensated by bulking agent like maltodextrin and polydextrose, and the later by adding a freezing point depressant like sorbitol

Au:

- The author changes of the sentences.

- Reduction of sucrose or fat to produce sucrose and fat free ice cream products affects the solids and the freezing point depress. The first can be compensated by bulking agent like maltodextrin and polydextrose, and the later by adding a freezing point depressant like sorbitol

The same for lines 143-144.

Sorbitol containing sugar-free products have a lower glycemic index and are about 60% more sweet than those containing sucrose (one gram gives 2.60 calories)’

Au:

- The author changes of the sentence.

- Sorbitol containing sugar-free products have a lower glycemic index and are about 60% more sweet than those containing sucrose (one gram gives 2.60 calories)’

The same for line 148:

Ice cream is frozen dairy product that includes healthy and nutritious aspects.

Au:

- The author changes of the sentence.

- Ice cream is frozen dairy product that includes healthy and nutritious aspects.

Materials and Methods

Line 164 (now 332): The information about freezing point calculation is not sufficient. Because you used thermometer to measure this parameter, it seems that you mean the temperature of ice cream when exited the ice cream machine. Hence, analytical description of this analysis is necessary.

Au:

- The author changes of the line.

- The freezing point was determined according to Marshall et al. [28] applying the digital thermometer. Where the freezing point of ice cream is measured as soon as it exits the ice cream machine by placing the thermometer in the treatment that comes out of the machine directly and recording the reading.

Line 340: The weight of melted ice cream expressed as % of initial weight of ice cream was measured according …

Au:

- The author changes of the line.

- The weight of melted ice cream expressed as % of initial weight of ice cream was measured according …

Results

The footnotes under each Table should be corrected accordingly as follows: Means with different superscript letters within the same row (or the same column) differ significantly (p <0.05).

Au:

- The author changes of the footnotes under each Table.

- Means with different superscript letters within the same row differ significantly (p < 0.05).
